# Intracellular Cl^−^ Regulation of Ciliary Beating in Ciliated Human Nasal Epithelial Cells: Frequency and Distance of Ciliary Beating Observed by High-Speed Video Microscopy

**DOI:** 10.3390/ijms21114052

**Published:** 2020-06-05

**Authors:** Makoto Yasuda, Taka-aki Inui, Shigeru Hirano, Shinji Asano, Tomonori Okazaki, Toshio Inui, Yoshinori Marunaka, Takashi Nakahari

**Affiliations:** 1Department of Otolaryngology-Head and Neck Surgery, Graduate School of Medical Science, Kyoto Prefectural University of Medicine, Kyoto 602-8566, Japan; inui1227@koto.kpu-m.ac.jp (T.-a.I.); hirano@koto.kpu-m.ac.jp (S.H.); 2Research Unit for Epithelial Physiology, Research Organization of Science and Technology, BKC, Ritsumeikan University, Kusatsu 525-8577, Japan; ashinji@ph.ritsumei.ac.jp (S.A.); t-inui@saisei-mirai.or.jp (T.I.); marunaka@koto.kpu-m.ac.jp (Y.M.); 3Department of Molecular Physiology, Faculty of Pharmaceutical Sciences, BKC, Ritsumeikan University, Kusatsu 525-8577, Japan; ph0034fe@ed.ritsumei.ac.jp; 4Saisei Mirai Clinics, Moriguchi 570-0012, Japan; 5Department of Molecular Cell Physiology, Graduate School of Medical Science, Kyoto Prefectural University of Medicine, Kyoto 602-8566, Japan; 6Research Institute for Clinical Physiology, Kyoto Industrial Health Association, Kyoto 604-8472, Japan

**Keywords:** nasal cilia, intracellular Cl^−^ concentration, Cl^−^ channels, CFTR, mucociliary clearance

## Abstract

Small inhaled particles, which are entrapped by the mucous layer that is maintained by mucous secretion via mucin exocytosis and fluid secretion, are removed from the nasal cavity by beating cilia. The functional activities of beating cilia are assessed by their frequency and the amplitude. Nasal ciliary beating is controlled by intracellular ions (Ca^2+^, H^+^ and Cl^−^), and is enhanced by a decreased concentration of intracellular Cl^−^ ([Cl^−^]_i_) in ciliated human nasal epithelial cells (cHNECs) in primary culture, which increases the ciliary beat amplitude. A novel method to measure both ciliary beat frequency (CBF) and ciliary beat distance (CBD, an index of ciliary beat amplitude) in cHNECs has been developed using high-speed video microscopy, which revealed that a decrease in [Cl^−^]_i_ increased CBD, but not CBF, and an increase in [Cl^−^]_i_ decreased both CBD and CBF. Thus, [Cl^−^]_i_ inhibits ciliary beating in cHNECs, suggesting that axonemal structures controlling CBD and CBF may have Cl^−^ sensors and be regulated by [Cl^−^]_i_. These observations indicate that the activation of Cl^−^ secretion stimulates ciliary beating (increased CBD) mediated via a decrease in [Cl^−^]_i_ in cHNECs. Thus, [Cl^−^]_i_ is critical for controlling ciliary beating in cHNECs. This review introduces the concept of Cl^−^ regulation of ciliary beating in cHNECs.

## 1. Introduction

The nasal and sinonasal epithelia are exposed to small inhaled airborne particles, such as allergens, chemicals, viruses and bacteria, which are removed via mucociliary clearance. Mucociliary clearance is a host defence mechanism of the respiratory tracts that comprises a thin mucous film and beating cilia [1,2,3,4,5]. In this process, small inhaled particles are entrapped by the thin mucous film (surface mucous layer) and swept away from the nasal cavity by the activity of the beating cilia. Mucociliary clearance can be compared with a conveyer-belt system that extrudes the small inhaled particles from airways; that is, the surface mucous layer is the belt and the beating cilia are the engine for driving mucociliary clearance. Thus, the beating cilia are essential to maintain healthy nasal and sinonasal mucosa and to prevent their dysfunction, such as in primary ciliary dyskinesia (PCD) or induced sinusitis [1,2,3,4,5,6]. Therefore, drugs stimulating ciliary beating are of particular importance to improve or prevent nasal and sinonasal diseases.

The functional activities of ciliary beating can be assessed by two parameters, ciliary beat frequency (CBF) and waveform [6,7,8,9,10,11,12,13,14]. Previous studies have shown that CBF is a key parameter controlling the rate of mucociliary clearance [6,7]. However, there is evidence that the waveform is also an important factor for assessing the functional activities of the cilia [8,9,10,11,12]. There are several parameters for the assessment of waveform [8,9,10,11,12], one of which is the amplitude of ciliary beating. Studies of airway and nasal ciliary cells have demonstrated that the amplitude of ciliary beating is an essential factor for ciliary transport [14,15,16]. An increase in ciliary bend angle (CBA), which is an index of ciliary beat amplitude, has been shown to enhance the transport of microbeads on the airway surface driven by beating cilia [14]. Moreover, various stimulations have increased CBA in addition to CBF [14,17,18,19,20,21]. Recently, we developed a novel method for measuring ciliary beat distance (CBD), another index of ciliary beat amplitude, using a planar sheet of ciliated human nasal epithelial cells (cHNECs) in primary culture [15,16,20,21].

The intracellular Cl^—^ concentration ([Cl^−^]_i_) has been demonstrated to regulate cellular functions in various cell types, such as Na^+^-permeable channels in foetal lung cells [22] and salivary ducts [23], Ca^2+^-regulated exocytosis and Ca^2+^-permeable channels in antral mucous cells [24], G-proteins [25], the cell cycle [26,27], tubulin polymerisation [28,29,30] and gene expression [31]. Moreover, a decrease in [Cl^−^]_i_ in airway ciliary cells, including cHNECs, has been shown to enhance CBD or CBA [15,16,20,21,32].

We recently demonstrated that daidzein and carbocisteine (CCis), which are activators of Cl^−^ channels, enhanced CBD mediated via a decrease in [Cl^−^]_i_ [15,16], suggesting that nasal secretion of Cl^−^ controls ciliary beating via [Cl^−^]_i_. At present, there are no known molecular targets of [Cl^−^]_i_ in airway ciliary cells or cHNECs. However, there is evidence that [Cl^−^]_i_ controls ciliary beating in cHNECs. In this review, we introduce the concept of Cl^−^ regulation of ciliary beating in cHNECs.

## 2. cHNECs

cHNECs isolated from nasal samples obtained during surgery or by brushing are cultured at the air–liquid interface (ALI) to differentiate into ciliated cells. cHNECs are widely used for cilia research [33,34,35,36,37]. Recent studies have demonstrated that cHNECs have some characteristic features that differ from ciliated airway cells of the trachea or lung. The ciliated tracheal and lung cells are sensitive to intracellular pH (pH_i_); a high pH_i_ increases CBF, and a low pH_i_ decreases CBF [15,16,20,21,38]. The pH_i_ can be changed by switching from a CO_2_/HCO_3_^−^-containing to a CO_2_/HCO_3_^−^-free solution, although the changes in pH_i_ are small. H^+^ is produced from CO_2_ by carbonic anhydrase. Therefore, the switch to a CO_2_/HCO_3_^−^-free solution increases pH_i_, and the return to a CO_2_/HCO_3_^−^-containing solution decreases it [15,16,20,21,38,39]. In tracheal ciliary cells, the switch to a CO_2_/HCO_3_^−^-free solution has been shown to increase CBF, mediated via an increase in pH_i_ [38]. However, in cHNECs, the switch to a CO_2_/HCO_3_^−^-free solution induced only a small and transient increase in pH_i_, leading to a slight or sometimes no increase in CBF [15,16,21]. We also applied the NH_4_^+^ pulse to change pH_i_. An application of the NH_4_^+^ pulse (an addition of NH_4_Cl, such as 25 mM) in extracellular fluid releases a small amount of NH_3_, which enters cells and traps H^+^ to produce NH_4_^+^ leading to an increase in pH_i_ [21,38,39], and it induced a larger increase in pH_i_ independent of CO_2_ than that induced by the CO_2_/HCO_3_^−^-free solution. In cHNECs, the NH_4_^+^ pulse induced transient CBF and pH_i_ increases. However, in the presence of acetazolamide (an inhibitor of carbonic anhydrase, which inhibits H^+^ production from CO_2_), it induced sustained increases, not transient, in pHi and CBF. These results indicate that cHNECs produce a large amount of H^+^, even under CO_2_/HCO_3_^−^-free conditions. A high level of H^+^ production suggests a high CO_2_ production or a high carbonic anhydrase activity in cHNECs.

Moreover, mouse nasal ciliary cells, unlike airway ciliary cells, have shown spontaneous CBF oscillation induced by periodic intracellular Ca^2^^+^ ([Ca^2^^+^]_i_) spikes and no increase in CBF during isobutylmethylxanthine (IBMX, an inhibitor of phosphodiesterase) stimulation [40]. Thus, cHNECs possess characteristic features distinct from tracheal or lung airway ciliary cells.

The characteristic features of cHNECs distinct from trachea and lung airway ciliary cells appear to be caused by the different embryological origins. The cHNECs are derived from the surface ectoderm, while ciliary cells of trachea and lung are from the endoderm. These characteristic features appear to be beneficial for cHNECs, which are exposed to air directly. In particular, a high H^+^ production in cHNECs prevents pH_i_ changes induced by fluctuations in CO_2_ concentrations during inspiration and expiration (0.3–40 mmHg). 

A decrease in [Cl^−^]_i_ increases the amplitude, CBA or CBD, but not CBF, in cHNECs [15,16,21], similar to airway ciliary cells [20]. These observations suggest that [Cl^−^]_i_ is an important ion involved in regulating ciliary beating in airway ciliary cells, including cHNECs.

## 3. Analysis of Ciliary Beating in cHNECs

Primary cHNEC cultures are grown at the ALI and form a cell sheet similar to nasal or sinonasal epithelia. The ciliary beating of cHNECs can be observed from the apical side using high-speed video microscopy (HSVM). Since CBA measurement is difficult in the apical view, a new parameter is required to assess the amplitude of ciliary beating. Recently, CBD has been proposed as a new parameter for assessing the amplitude using HSVM and an image analysis programme.

### 3.1. HSVM

Recent developments in HSVM have allowed the observation of fine movements of airway ciliary beating, the analysis of which has enabled the measurement of the functional parameters of ciliary beating, not only the CBF but also the waveform or the beating pattern, including amplitude [9,10,11]. HSVM has been shown to be an effective tool for PCD diagnosis [11,41].

CBF has been established as the functional parameter of ciliary beating. However, there is no standard definition or consistency in the evaluation of ciliary waveforms, and the quantitative assessment of the waveform or of the pattern of ciliary beating remains controversial because of the waveform’s complexity [11]. In our studies, the amplitude measured as CBD or CBA was shown to be an important factor in stimulating airway ciliary transport [4,5,6,7,8,9,10,11,12] and is proposed as a parameter for assessing the ciliary waveform.

Fine images of beating cilia with a high resolution in space and time are essential to measure the frequency and amplitude, and HSVM is a useful tool for this purpose. To observe ciliary beating, an inverted microscope with phase-contrast or differential interference contrast (DIC) optics as well as a high magnification objective lens (i.e., 60× or 100×) are suitable. However, the level of magnification should be selected according to the experimental purpose. The beating cilia move at a depth 5–20 µm above the apical surface of ciliary cells, and the focal plane depth of a high magnification lens with phase-contrast or DIC optics is <5 µm. In our experiments, we sometimes used HSVM equipped with a 60× objective lens without DIC or phase-contrast equipment for observing whole ciliary movements. Moreover, considering that a microscope light source may occasionally be insufficient for HSVM, especially when using DIC optics, a suitable optic, such as phase-contrast, was required. Given that CBF is highly temperature-dependent [3], a temperature-controlled micro-perfusion chamber was also necessary.

HSVM allows beating cilia from isolated ciliary cells to be viewed from three directions—a sideways profile, beating toward the lens (vertical direction), and from above [12]. However, in a sheet of cultured cells, such as cHNECs, most cells are viewed from the apical side (Figure 1A,B). CBF can be measured in images recorded from all directions. However, it is difficult to measure CBD using the side view or vertical images, but an apical image is suitable for CBD measurement (Figure 1 and Figure 2).

### 3.2. Digital High-Speed Camera

Given their high frequency of 8–25 Hz at 37 °C, it is difficult to observe the fine movements of beating airway cilia at the National Television System Committee frame rate (30 fps). To visualise the complete cycle of ciliary beating, a high video recording rate (i.e., 500 Hz) is essential.

### 3.3. CBF Measurement

Available image analysis software for CBF measurement has been based on light intensity changes in the pixels of the recorded images over time (Figure 1). A semiautomated programme can determine CBF in selected lines of measured cilia [11]. At present, sample movements during perfusion occasionally introduce artefacts in the CBF obtained from the programme. Thus, while semiautomated CBF analysis has some advantages, such as a shorter time requirement, certain limitations are also present. When calculating CBF, light intensity peaks in the image obtained from the programme are counted (Figure 1 and Figure 2).

### 3.4. CBD Measurement

Ciliary beating is coordinated with metachronal waves, which exhibit a whip-like movement. In the effective stroke of ciliary beating, the cilium tip makes an arc with a maximum speed in the plane perpendicular to the cell surface, while in the recovery stroke, the cilium swings back close to the cell surface (Figure 2). Studies have demonstrated that an abnormal waveform of ciliary beating induces ciliary dysfunction, leading to PCD, which is characterised by situs inversus totalis, sinusitis, and bronchiectasis [29]. The quantitative evaluation of ciliary waveform or ciliary wave pattern has been proposed to diagnose PCD based on an abnormal ciliary waveform or ciliary beat pattern [11,41]. Among the waveform evaluation parameters, we previously used CBD or CBA as an index of ciliary beat amplitude [4,5,6,7,8,9,10,24]. CBD has been characterised as the distance between the maximum forward and backward movements of the cilia tip [8,9,10,24,30,42] and CBA as the angle between the start and the end cilium positions in the effective stroke [4,5,6,7,8,24]. As shown in Figure 1 and Figure 2, CBD and CBA are indices of ciliary beating amplitude and are thus the parameters for evaluating the ciliary beating waveform. It is certain that CBD or CBA are, at least, two of the most important parameters for evaluating the waveform of ciliary beating. Previous studies have demonstrated that an increase in CBA or CBD enhanced the transport of microbeads in the airway surface [4] or on the surface of cHNECs [9,10,24].

Moreover, in some cases, increases in CBD or CBA occurred without any increase in CBF. As mentioned above, a CBD increase alone enhanced microbead transport [9,10,24], although increases in CBF can also have this effect. Thus, CBD measurement is essential for evaluating the ciliary function in addition to CBF measurement.

Although various abnormal waveforms have been reported in PCD, the amplitude of ciliary beating, CBD and CBA, remain controversial as a parameter for PCD diagnosis [11]. However, CBD and CBA are important factors for evaluating normal airway cilia activity because an increase in CBD or CBA was shown to enhance microbead transport [14,15,16].

## 4. Changes in [Cl^−^]_i_

[Cl^−^]_i_ has been shown to modulate cellular functions in many cell types [22,23,24,25,26,27,28,29,30,31,32] and to be affected by cell volume changes. Many agonists activating ion transport, such as Cl^−^ secretion and K^+^ release, have been demonstrated to evoke cell shrinkage under the isosmotic condition in many cell types [20,22,43,44,45,46]. This occurs when an increase in [Ca^2+^]_i_ or cyclic adenosine monophosphate accumulation stimulated by an agonist activates K^+^ and Cl^−^ channels, leading to the cellular release of KCl [20,22,32,46]. The KCl release generates an osmotic gradient between the intracellular and extracellular space and is followed by a fluid efflux. Finally, the cell volume decreases to an equilibrium condition (isosmotic cell shrinkage). In airway epithelia, the activation of Cl^−^ secretion (Cl^−^ release from cells) also accompanies K^+^ release for the maintenance of the intracellular electroneutrality. The KCl release, which generates a hypoosmotic condition in intracellular space, induces fluid efflux to evoke cell shrinkage [46]. Agonists, such as procaterol, CCis and daidzein, have already been shown to stimulate cell shrinkage in airway ciliary cells and cHNECs [15,16,20,21]. Furthermore, this isosmotic cell shrinkage has been demonstrated to decrease [Cl^−^]_i_ in airway ciliary cells and cHNECs [15,16,21].

In general, K^+^ and Cl^−^, which are the main intracellular cation and anion, respectively, are membrane-permeable ions, because the cell membrane has ion transporters and channels for K^+^ and Cl^−^. The isosmotic cell shrinkage, which is caused by the KCl release, decreases [Cl^−^]_i_. Figure 3 shows the mechanism for decreasing [Cl^−^]_i_ induced by cell shrinkage using a model cell for the calculation of [K^+^]_i_ and [Cl^−^]_i_. In this model cell, we assume that intracellular K^+^ concentration ([K^+^]_i_) and [Cl^−^]_i_ are 125 and 45 mM in unstimulated condition, respectively. Assuming that an agonist decreases cell volume by 20%, the [K^+^]_i_ is maintained at 125 mM, but [Cl^−^]_i_ decreases from 45 to 25 mM, as shown in Figure 3. Thus, in this model cell, the isosmotic cell shrinkage decreases [Cl^−^]_i_ without any change in [K^+^]_i_ (Figure 3) [46]. The calculation in this model cell indicates that the [Cl^−^]_i_ decrease occurs during cell shrinkage under physiological conditions.

Daidzein is an agonist that stimulates Cl^−^ channels. It has been demonstrated to decrease only [Cl^−^]_i_, coupled with cell shrinkage, in cHNECs (Figure 4). This can be monitored using N-(ethoxycarbonylmethyl)-6-methoxyquinolinium bromide (MQAE), a Cl^−^ sensitive fluorescent dye. A decrease in [Cl^−^]_i_ increases the MQAE fluorescence intensity, whereas an increase in [Cl^−^]_i_ decreases it (Figure 4A,B) [15,16,20,21,47]. Daidzein, an agonist that stimulates Cl^−^ channels, has been shown to induce an [Cl^−^]_i_ decrease in cHNECs [15]. Figure 4A,B, in which cellular shapes have been superimposed on the MQAE fluorescence images, show changes in the cellular outline and MQAE fluorescence before and after daidzein stimulation. Daidzein stimulates cell shrinkage and increases the fluorescence intensity of MQAE. Relative changes in cell volume and MQAE fluorescence ratio (F_0_/F) are shown in Figure 4C, where subscript “0” refers to the time when the stimulation was initiated. Accordingly, daidzein decreased cell volume to 81% and F_0_/F to 78%. In cHNECs, a decrease in [Cl^−^]_i_ (i.e., decreased cell volume) increased only CBD but not CBF (Figure 4D).

## 5. [Cl^−^]_i_ Regulation of Ciliary Beating in cHNECs

### 5.1. Effects of Decreased [Cl^−^]_i_ on CBF and CBD

The effects of decreased [Cl^−^]_i_ on CBF and CBD are presented in Figure 5. A switch to a CO_2_/HCO_3_^−^-free solution alone decreased [Cl^−^]_i_ [15,16,21]. This [Cl^−^]_i_ decrease is explained by the inhibition of NaCl entry via Na^+^-HCO_3_^−^ cotransporter (NBC) and Cl^−^/HCO_3_^−^ exchange (anion exchange, AE) in the CO_2_/HCO_3_^−^-free solution. Decreases in Cl^−^ entry by NBC and AE inhibition decreases [Cl^−^]_i_ and also increases CBD significantly, but not CBF. Experiments were performed under CO_2_/HCO_3_^−^-free conditions.

There are several methods to decrease [Cl^−^]_I_, including the substitution of Cl^−^ in the perfusion solution, inhibition of Cl^−^ entry and activation of Cl^−^ efflux. For Cl^−^ substitution, NO_3_^−^ was used instead of Cl^−^ in the perfusion solution (Figure 5A). Decreasing extracellular Cl^−^ concentration ([Cl^−^]_o_) promoted a decrease in F_0_/F in cHNECs, indicating a decrease in [Cl^−^]_i_. Values of F_0_/F decreased according to [Cl^−^]_o_ decrease. In this experiment, F_0_/Fs were set at four levels, dependent on [Cl^−^]_o_ in cHNECs, revealing that a decrease in [Cl^−^]_o_ decreases [Cl^−^]_i_. A decrease in [Cl^−^]_i_ with the application of NO_3_^−^ solution increased CBD, but not CBF. Increases in CBD were dependent on decreases in [Cl^−^]_i_; a large decrease in [Cl^−^]_i_ induced a large increase in CBD (Figure 5A).

The effects of bumetanide (bum, an inhibitor of Na+/K+/2Cl- cotransporter (NKCC)), daidzein (an activator of Cl^-^ channels) and carbocisteine (CCis, an activator of Cl^−^ channels including cystic fibrosis transmembrane conductance regulator (CFTR)) on CBF and CBD under CO_2_/HCO_3_^−^-free conditions are shown in Figure 5B. The switch to CO_2_/HCO_3_^−^-free solution decreased [Cl^−^]_i_ as explained previously. Bum, which inhibits Na^+^, K^+^ and 2Cl^−^ entry, decreased F_0_/F ([Cl^−^]_i_) (Figure 5B) [16,21]. An activation of Cl^−^ channels by daidzein or CCis also decreased [Cl^−^]_i_ (Figure 5B) [15,16]. Decreases in [Cl^−^]_i_ induced by NO_3_^−^ solution, bum, daidzein and CCis increased CBD but not CBF.

### 5.2. Effects of Increase in [Cl^−^]_i_ on CBF and CBD

To increase [Cl^−^]_i_, the Cl^−^ channel inhibitor 5-nitro-2-(3-phenylpropylamino)benzoic acid (NPPB) was used. NPPB inhibits Cl^−^ release from cHNECs, leading to an increase in [Cl^−^]_i_. The addition of NPPB increased [Cl^−^]_i_ by inhibiting Cl^−^ release in cHNECs (Figure 5C). A similar increase in [Cl^−^]_i_ was induced by adding a cystic fibrosis transmembrane conductance regulator inhibitor (CFTR(inh)-172) to cHNECs [15]. Unlike a decrease in [Cl^−^]_i_, NPPB- and CFTR(inh)-172-induced increases in [Cl^−^]_i_ decreased both CBF and CBD. Thus, the range of [Cl^−^]_i_ activating CBF differed from that activating CBD.

### 5.3. Effects of Decrease in [Cl^−^]_i_ on CBF and CBD at 25 °C

A decrease in [Cl^−^]_i_ increased CBD, but not CBF, and an increase in [Cl^−^]_i_ decreased both CBF and CBD. However, a number of reports have suggested that a decrease in [Cl^−^]_i_ increases CBF [32,48,49,50]. These studies were performed at room temperature. The effects of decreased [Cl^−^]_i_ on CBF and CBA were examined at 25 °C (Figure 5D). The application of Cl^−^-free (NO_3_^−^) solution, bum or daidzein at 25 °C increased both CBF and CBD. These results suggest that a decrease in [Cl^−^]_i_ enhances both CBF and CBD, although the extent of the increases in CBF and CBD differed according to the temperature [16,20]. These results appear to suggest that [Cl^−^]_i_ may inhibit CBF and CBD and that the Cl^−^ concentration–response curve of CBF may shift to a higher concentration than that of CBD, indicating that the [Cl^−^]_i_ affects both CBF and CBD. The Cl^−^-binding affinity for the structures controlling CBD and CBF may depend on different temperatures.

### 5.4. Effects of Increased CBD on the Microbead Movement in cHNECs

To examine the effects of an increase in CBD on ciliary transport, microbeads were applied to cHNECs [14,15,16]. Microbeads reaching the surface of cHNECs were moved by the surface fluid flow driven by beating cilia. Microbead movement was observed using HSVM (60 fps). Figure 6A shows the movement of a microbead that reached the surface of a cHNEC (before daidzein addition). The large arrows show the positions of a microbead reaching the cell surface, while small arrows show the distance moved in 33 ms (over two frames). Figure 6B shows video images 5 min after daidzein addition, which enhanced the distance that a microbead moved. Figure 6C shows the CBD ratio and microbead movement before and after the daidzein addition. Accordingly, the increase in CBD enhanced microbead movement. Figure 6D shows the CBD ratio and microbead movement before and after CCis addition. Similarly, CCis addition enhanced CBD and microbead movement. CCis has already been shown to increase only CBD, without increasing CBF. The aforementioned results show that an increase in CBD alone enhances microbead movement, suggesting that CBD, in addition to CBF, is an important parameter for assessing the functions of ciliary beating.

## 6. [Cl^−^]_i_ Modulation of CBF and CBD in cHNECs

Ciliary beating is maintained by force generated by molecular motors called dyneins [51]. Motile cilia can have two functionally distinct dyneins, outer dynein arms (ODAs), which control frequency, and inner dynein arms (IDAs), which control waveform, including CBD [52,53]. This suggests that a decrease in [Cl^−^]_i_ stimulates both ODA and IDA activity, whereas an increase in [Cl^−^]_i_ inhibits both. However, an [Cl^−^]_i_ decrease stimulates ODAs to increase CBF at 25 °C, but not at 37 °C. Based on these observations, axonemal structures controlling ODAs and IDAs might have a sensor to which [Cl^−^]_i_ binds. The binding of Cl^−^ to this sensor would decrease ODA and IDA activities, consequently decreasing CBF and CBD. The Cl^−^ concentration–response curve for CBF might shift to a higher concentration than that for CBD. The Cl^−^-binding affinity for the structures controlling the ODAs might become decreased at a low temperature, and if no Cl^−^ binds to the structures, ODA and CBF activity may increase. The Cl^−^-binding affinity for the structures controlling CBF might be higher than that controlling CBD.

There are numerous reports showing that [Cl^−^]_i_ modulates cellular functions in many cell types [22,23,24,25,26,27,31,54]. These observations suggest that cells have chloride sensors transducing this signal to the targets.

Piala et al. (2014) showed that [Cl^−^]_i_ binds to the kinase domain of with-no-lysine kinase (WNK) 1 and inhibits WNK1 autophosphorylation, inhibiting its activity [55]. WNK1–4 are serine/threonine protein kinases lacking a lysine residue within subdomain II of the kinase domain. They share a common structure with >80% identity in their kinase domain [56,57], indicating that the chloride-binding site is conserved among the WNKs. Kinase studies have demonstrated that the affinities for Cl^−^ differ among WNK1, WNK3, and WNK4 [57,58]. WNK4 activity is inhibited when [Cl^−^]_i_ is 0–40 mM, while those of WNK1 and WNK3 become inhibited at 60–150 mM and 100–150 mM, respectively [57]. WNK1 and WNK4 have been shown to regulate CFTR [59] and NKCC1 of olfactory sensory cilia [60]. WNK4 has also been shown to regulate epithelial Na^+^ channels (ENaC) in various tissues including airway [61,62]. CFTR and ENaC exist in nasal epithelia [15,16,63]. Moreover, WNK1 and WNK4 have been demonstrated to exist in olfactory sensory cilia [60]. Although the presence of WNK1 and WNK4 in motile cilia of cHNECs still remains uncertain, these observations may suggest that WNK1 and WNK4 are physiological [Cl^−^]_i_ sensors in cHNECs, as shown in an NCC study using WNK4 K/O mice [64]. Further studies are necessary.

WNK actions have been extensively studied in NaCl cotransporters (NCCs) of distal nephrons. WNK4 has been shown to inhibit epithelial Na^+^ channels [61]. WNK1 and WNK4, which phosphorylate OSR1 (oxidative stress-responsive kinase 1) and SPAK (sterile20-related proline-, alanine-rich kinase), activate NKCC [65,66]. The inhibition of WNK1, WNK3 and WNK4, which depends on their affinities for Cl^−^, have been shown to inhibit NCCs [66,67]. Thus, although WNK1, WNK3 and WNK4 are inhibited by Cl^−^, WNK1 and WNK4 may be active in the physiological range of [Cl^−^]_I_ in many cell types, including cHNECs.

To examine the effects of [Cl^−^]_i_ on CBD and CBF in cHNECs, the relative changes in CBD and CBF (CBD ratio and CBF ratio), which were reported in previous studies [15,16,21], were plotted against the MQAE fluorescence ratio (F_0_/F, an index of [Cl^−^]_i_) (Figure 7). CBD (red marks) was inhibited at a lower [Cl^−^]_i_ than CBF (blue marks). The inhibition of CBD and CBF by increasing [Cl^−^]_i_ seems to be similar to the inhibition of WNK4 and WNK1 obtained by increasing [Cl^−^]_i_ in the previous studies in vitro [55,57,58].

These studies may suggest that WNK4 exists in axonemal structures controlling IDAs and WNK1 exists in those controlling ODAs. The different subtypes of WNK, such as WNK4 or WNK1, may induce different actions in response to changes in [Cl^−^]_i_. Unfortunately, whether the axonemal structures controlling IDA and ODA possess WNK4 and WNK1 and the exact [Cl^−^]_i_ in cilia of cHNECs is unknown. If WNK4 and WNK1 exist in the axonemal structure controlling IDA and ODA, respectively, the effects of [Cl^−^]_i_ on CBF and CBA may be explained. For example, an [Cl^−^]_i_ decrease stimulated by cell shrinkage may activate WNK4, leading to the activation of IDA-controlling structures (CBD increase), but not WNK1, resulting in the continuing activation of the ODA-controlling structures (no CBF increase). A low temperature, such as room temperature, may decrease the affinities of WNK4 and WNK1 for Cl^−^, leading to both WNK4 and WNK1 activation, which may increase CBD and CBF.

PCD patients have mutations in microtubule motor proteins and other structural proteins controlling ciliary beating. These proteins may have Cl^−^ sensors, such as WNK1and WNK4, and their activities may be affected by [Cl^−^]_i_. Their mutation may abolish activations of CBD and CBF in response to a decrease in [Cl^−^]_i_ and may make ciliary activities worse. Further studies are required.

A previous study showed that a decrease in [Cl^−^]_i_ increased the dynein affinity for microtubules [68]. Fluctuations in [Cl^−^]_i_, which modulate G proteins and GTPase [28,30,69], may also affect the interactions between tubulin and dyneins via G proteins and GTPase, which may change the sliding speed or force generation [70], leading to changes in CBF and CBD. These actions may also be mediated by WNK4 and WNK1 or be independent from them.

## 7. Conclusions

[Cl^−^]_i_ is an important signal ion that modulates ciliary beating in cHNECs. Nasal epithelia in primary culture have already shown to secrete Cl^−^ [63,71], although nasal mucosal glands including goblet cells also secrete Cl^-^ and mucins maintaining the nasal mucous layer. Under physiological conditions, Cl^-^ secretion in nasal epithelia maintains the surface serous fluid layer just below the mucous film, in which the cilia beat. Thus, Cl^−^ secretion in cHNECs maintains the mucociliary clearance in nasal and sinonasal epithelia. Our studies in cHNECs demonstrated that an activation of Cl^−^ secretion induces a decrease in [Cl^−^]_i_, which increases CBD. Moreover, airway cilia express CFTR [20]. Microdomains in the cilia are suggested to be isolated from the cell body, and their circumstances may be different from those of the cell body [17]. This may suggest that the activation or inhibition of CFTR induces a larger change in [Cl^−^]_i_ in the cilia than in the cell body. The summary of Cl^-^ regulation of ciliary beating in cHNECs is shown in Figure 8. 

The drugs stimulating Cl^−^ secretion, such as CCis, improve the symptoms of nasal and sinonasal diseases by increasing CBD. Moreover, a decrease in [Cl^−^]_i_ may be a new therapeutic tool for improving nasal and sinonasal problems. However, the [Cl^−^]_i_ sensor of cHNECs remains unknown. We speculate WNK1 and WNK4 as candidates of the [Cl^−^]_i_ sensor controlling CBF and CBD. Nevertheless, the response of other kinases to an [Cl^−^]_i_ decrease must not be neglected. Further studies should provide elucidation.

## Figures and Tables

**Figure 1 ijms-21-04052-f001:**
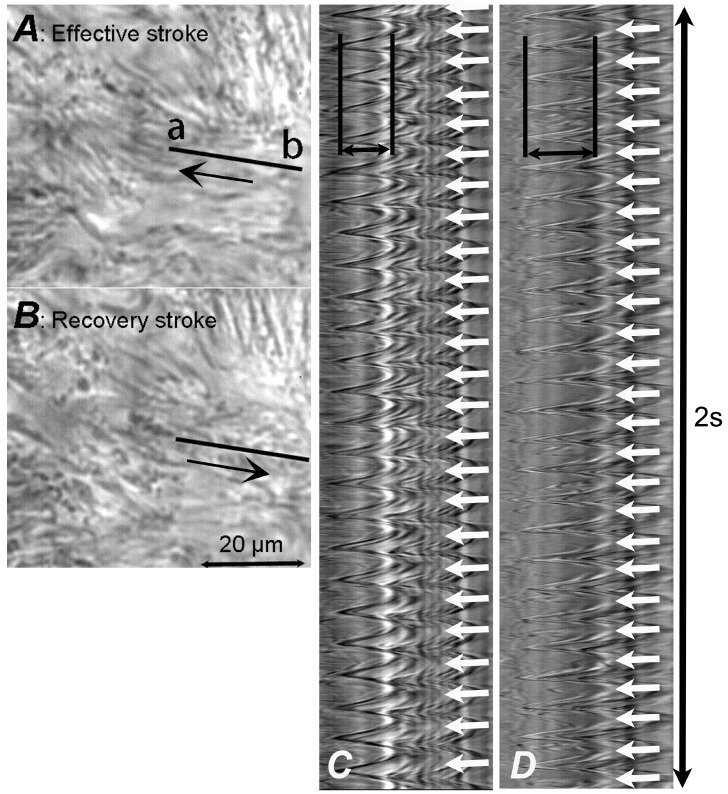
Video frame images of ciliated human nasal epithelial cells (cHNECs) recorded using high-speed video microscopy (500 fps) and images showing changes in the light intensity returned from the image analysis programme. (**A**,**B**) Video frame images of cHNECs. Panel A shows cilia in the end position of the effective stroke and the arrow shows the direction of the effective stroke, and Panel B shows cilia in the end position of the recovery stroke and the arrow shows the direction of the recovery stroke. (**C**) The light intensity changes for 2 s on line a–b marked in panel A. The distance between two lines marked by ←→ represents the ciliary beat distance (CBD), while the number of peaks marked by ↓ represents ciliary beat frequency (CBF, 12.5 Hz). (**D**) Light intensity changes of the ciliary beating in cHNECs during stimulation with 100 µM daidzein (13 Hz). The line was placed on the same position of a cHNEC as shown in panels A and B. The traces of light intensity changes clearly show that daidzein increases CBD, but not CBF.

**Figure 2 ijms-21-04052-f002:**
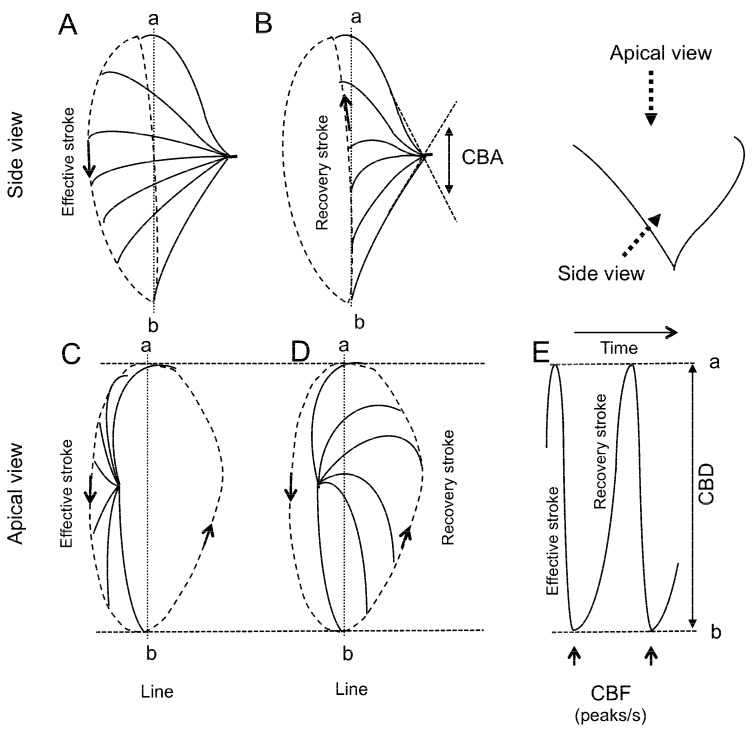
Schematic diagram of ciliary beating. The cilium travels in different planes during the effective and recovery strokes. During the effective stroke, the cilium tip moves in an arc with maximum speed in the plane perpendicular to the cell surface, and then, in the recovery stroke, the cilium swings back close to the cell surface. (**A**,**B**) Side view. Panels A and B show the effective stroke and the recovery stroke, respectively. The angle between the start and the end of the effective stroke is the ciliary beat angle (CBA, an index of ciliary beat amplitude). (**C**,**D**) Apical view. Panels C and D show the effective stroke and the recovery stroke, respectively. (**E**) The time course of changes in the light intensity of line a–b. To assess the amplitude of ciliary beating in the apical view, the distance between the start and the end of the effective stroke was measured. The measured distance is the ciliary beat distance (CBD, another index of ciliary beat amplitude). CBF, ciliary beat frequency.

**Figure 3 ijms-21-04052-f003:**
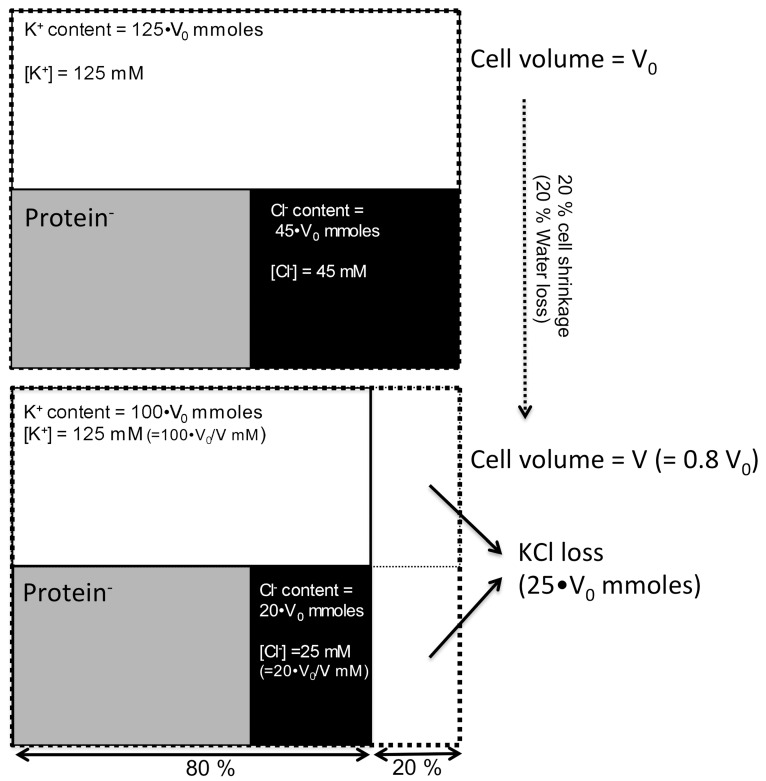
Decrease in [Cl^−^]_i_ induced by cell shrinkage under an isosmotic condition. In a model cell, [K^+^]_i_ and [Cl^−^]_i_ are 125 and 45 mM, respectively, in the unstimulated condition, while impermeable anions, such as proteins, maintain electroneutrality. Agonist stimulation activates K^+^ or Cl^−^ channels or both, leading to KCl release. Assuming an agonist decreases cell volume by 20% (KCl release decreases cell volume from V_0_ to V (0.8•V_0_) under isosmotic conditions), the amount of KCl loss (mM) is 25•V_0_. The KCl release from the cell does not alter [K^+^]_i_, and it is maintained at 125 mM, but [Cl^−^]_i_ decreases to 25 mM. Thus, under isosmotic conditions, the cell shrinkage decreases only [Cl^−^]_i_ while maintaining [K^+^]_i_.

**Figure 4 ijms-21-04052-f004:**
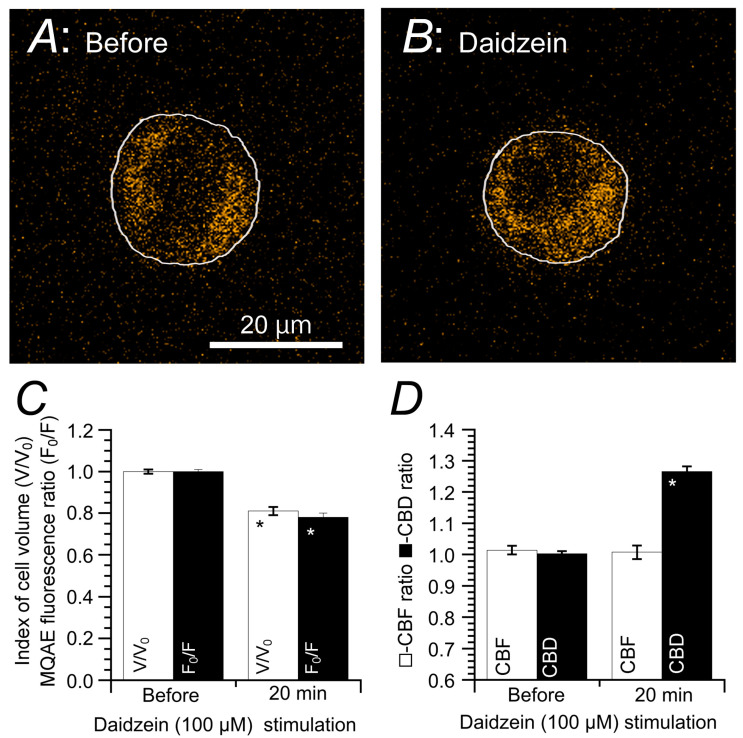
Changes in [Cl^−^]_i_ and cell volume following daidzein stimulation. Prior to daidzein (100 µM) stimulation, ciliated human nasal epithelial cells (cHNECs) were first perfused with a CO_2_/HCO_3_^−^-containing control solution for 5 min, followed by a CO_2_/HCO_3_^−^-free control solution for another 5 min. Then, a cell outline was superimposed on the N-ethoxycarbonylmethyl-6-methoxyquinolinium bromide (MQAE) fluorescence. (**A**) The MQAE fluorescent image of a cHNEC immediately prior to daidzein stimulation. (**B**) The MQAE fluorescent image of a cHNEC 20 min after daidzein stimulation (100 μM). Daidzein stimulation decreased cell volume and enhanced MQAE fluorescence intensity, indicating that daidzein decreased [Cl^−^]_i_ coupled with cell shrinkage. (**C**) Changes in MQAE fluorescence and cell volume. Daidzein decreased cell volume (V/V_0_), leading to decreased [Cl^−^]_i_. (**D**) Changes in CBF and CBD during daidzein stimulation. Daidzein stimulation increased CBD but not CBF. * indicates results significantly different from the control values (*p* < 0.05). Error bars indicate SE. This figure was modified from Inui et al., 2018 [16] with permission.

**Figure 5 ijms-21-04052-f005:**
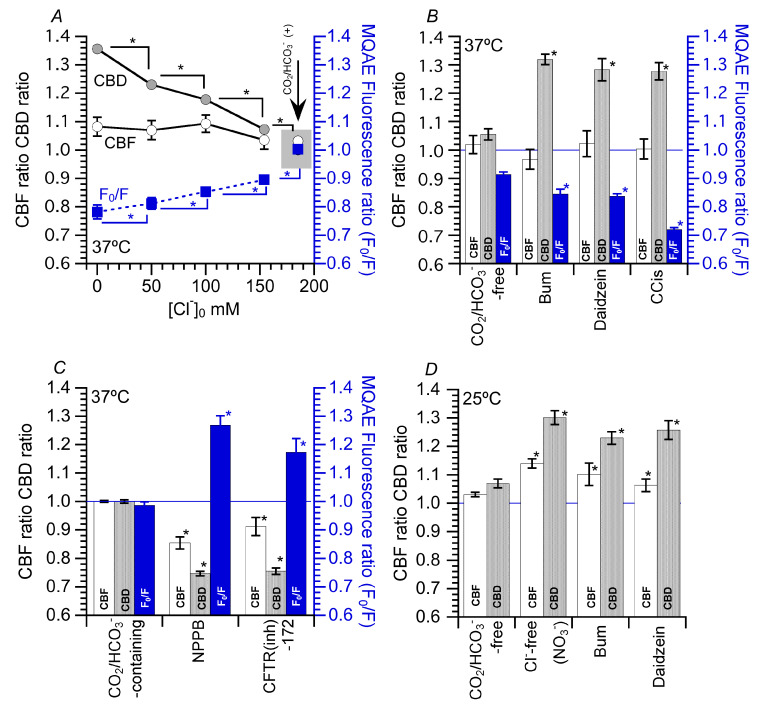
Effects of decreased intracellular Cl^−^ ([Cl^−^]_i_) on ciliary beat frequency (CBF) and ciliary beat distance (CBD). (**A**) Effects of Cl^−^-free (NO_3_^−^) solution on CBF, CBD and fluorescence ratio (F_0_/F) of N-ethoxycarbonylmethyl-6-methoxyquinolinium bromide (MQAE). First, the perfusion solution was switched from a CO_2_/HCO_3_^−^-containing solution to a CO_2_/HCO_3_^−^-free solution. This switch decreased F_0_/F (decrease in [Cl^−^]_i_) due to Na^+^-HCO_3_^−^ cotransporter and Cl^−^/HCO_3_^−^ exchange inhibition. The decreasing extracellular Cl^−^ concentration ([Cl^−^]_o_) decreased [Cl^−^]_i_. The decreased levels of F_0_/F were dependent on [Cl^−^]_o_. The decreases in [Cl^−^]_i_ increased CBD, the levels of which are dependent on [Cl^−^]_i_. However, a decreased [Cl^−^]_i_ did not increase CBF. (**B**) Effects of bumetanide (bum), daidzein and carbocisteine (CCis) on CBF, CBD and F_0_/F ([Cl-]_i_). Experiments were performed in a CO_2_/HCO_3_^−^-free solution. Bum, daidzein and CCis decreased F_0_/F, indicating decreases in [Cl^−^]_i_. These decreases in [Cl^−^]_i_ increased CBD, but CBF remained unchanged. (**C**) Effects of Cl^−^ channel blockers on CBF, CBD and F_0_/F ([Cl^−^]_i_). 5-nitro-2-(3-phenylpropylamino)benzoic acid (NPPB) and a cystic fibrosis transmembrane conductance regulator inhibitor (CFTR(inh)-172) increased F_0_/F ([Cl^−^]_i_). Increases in [Cl^−^]_i_ (F_0_/F) decreased both CBF and CBD. (**D**) Effects of Cl^−^-free (NO_3_^−^) solution, bum, daidzein and CCis on CBF and CBD at 25 °C. Unlike at 37 °C, Cl^−^-free (NO_3_^−^) solution, bum, daidzein and CCis increased both CBF and CBD. Thus, a decrease in [Cl^−^]_i_ enhances both CBF and CBD at 25 °C. * indicates results significantly different from the control values (*p* < 0.05). Error bars indicate SE.

**Figure 6 ijms-21-04052-f006:**
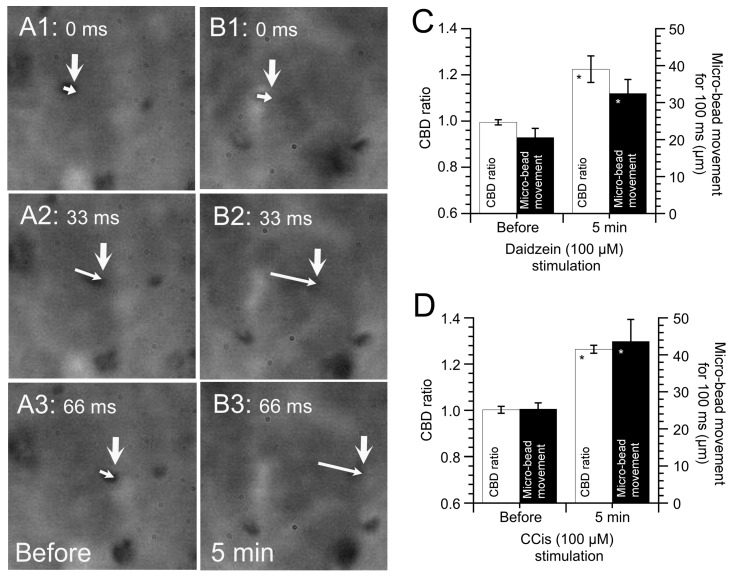
Latex microbead movement driven by ciliated human nasal epithelial cells. Panels A and B show six consecutive images recorded every 33 ms. (**A**) Before daidzein addition. (**B**) Five min after daidzein (100 µM) addition. The large arrows indicate the initial position of a microbead, and the small arrows indicate the distance that the microbead has moved. Daidzein stimulation enhanced microbead movement. (**C**) Daidzein-induced enhancement of ciliary beat distance (CBD) and microbead movement. Daidzein stimulation enhanced CBD and microbead movements. (**D**) Carbocisteine (CCis)-induced enhancement of CBD and microbead movements. CCis stimulation enhanced CBD and microbead movements. * indicates results significantly different from the control values (*p* < 0.05). Error bars indicate SE.

**Figure 7 ijms-21-04052-f007:**
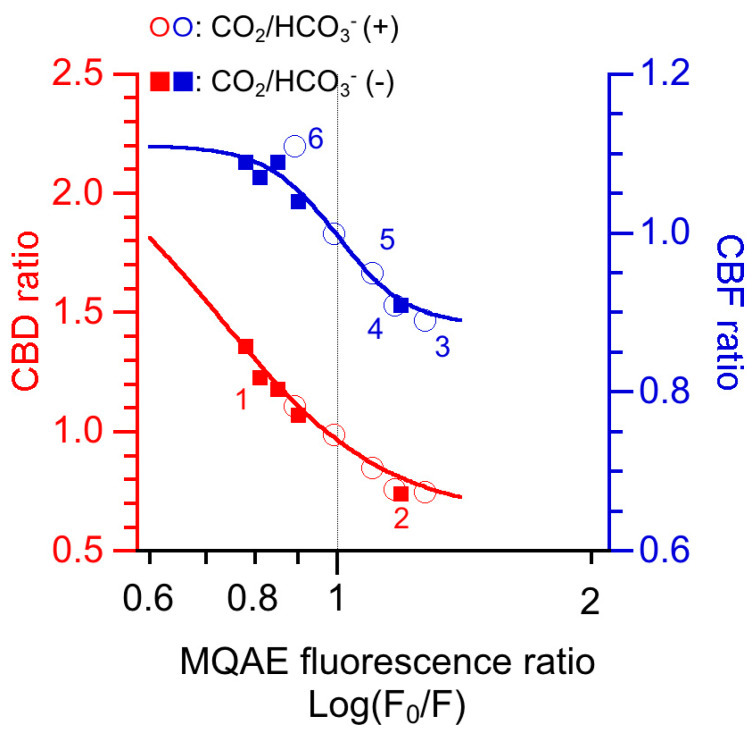
The effects of intracellular Cl^−^ ([Cl^−^]_i_) on relative changes in the ciliary beat distance (CBD) and ciliary beat frequency (CBF) (CBD ratio and CBF ratio) in cHNECs. {Cl^−^}_i_s are monitored using the N-(ethoxycarbonylmethyl)-6-methoxyquinolinium bromide (MQAE) fluorescence ratio (F_0_/F). F_0_/F is an index of [Cl^−^]_i_, not exact [Cl^−^]_i_. In this figure, the F_0_/F, CBD ratios and CBF ratios have been published in previous experiments performed in both the presence and the absence of CO_2_/HCO_3_^−^ [15,16,21]. The closed squares labelled 1 were obtained from the Cl^−^-free (NO_3_^−^) solution experiments, and those labelled 2 were obtained from the 5-nitro-2-(3-phenylpropylamino)benzoic acid (NPPB) experiments. The open circles 3 and 4 were obtained from the NPPB and cystic fibrosis transmembrane conductance regulator inhibitor-172 experiments, respectively. The open circles 5 and 6 were obtained from the T16Ainh-A01 (a Ca^2+^-activated Cl^−^ channel inhibitor) and the T16Ainh-A01 plus carbocisteine experiments, respectively. CBD was inhibited at a lower [Cl^−^]_i_ than CBF.

**Figure 8 ijms-21-04052-f008:**
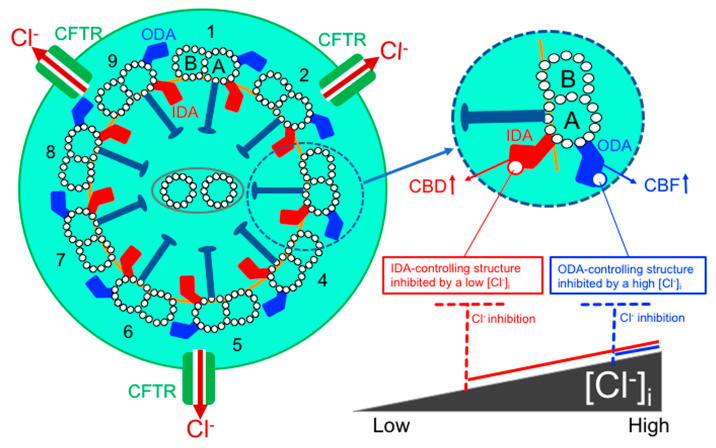
Cl^−^ modulation of ciliary beating in cHNECs. Ciliary membrane express CFTR [20]. Inner dynein arms (IDAs) control waveform including CBD and outer dynein arms (ODAs) control CBF. Axonemal structures controlling IDAs and ODAs appear to have Cl^−^ sensors. The sensors binding Cl^−^ may inhibit the axonemal structures controlling IDAs or ODAs leading to CBD decrease and CBF decrease. [Cl^−^]_i_ inhibiting IDA-controlling structures may be lower than that inhibiting ODA-controlling structures. Under the resting condition, ODAs are activated and IDAs are partly inactivated. At a lower [Cl^−^]_i_, ODAs continue to be active and IDAs are activated. At a higher [Cl^−^]_i,_ both ODAs and IDAs are inactivated.

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
