# Peer review of "Intracellular Cl Regulation of Ciliary Beating in Ciliated Human Nasal Epithelial Cells: Frequency and Distance of Ciliary Beating Observed by High-Speed Video Microscopy"

_ijms, 2020, doi:10.3390/ijms21114052_

Round 1

Reviewer 1 Report

All comments are also included in the pdf file.

The review by Yasuda et al., presents current knowledge on motile cilia beating regulation by intrinsic levels of chloride anion. Presented work summarize mainly achievements of Authors themselves. On the other hand, there are not many other studies in this field. Generally, the manuscript comprises an intriguing topic that would be of interest to readers of IJMS Special Issue on "Molecular Researches on Cilia" and therefore I recommend this manuscript for the publication.

However, before publication I would like to ask Authors to address several issues to improve the manuscript, making the presented ideas more accessible for readers that are not familiar with ion balance within the cells.

Below, find my comments that hopefully will help to improve the manuscript:

Manuscript main body:

cHNECs:

“However, in cHNECs, the switch to a CO2/HCO3−-free solution induced only a small and transient increase in pHi, leading to a slight or sometimes no increase in CBF [15,16,21]. An NH4+ pulse, which induces a large increase in pHi independent of CO2 [21,38,39], significantly increased CBF and pHi in cHNECs, unlike the switch to a CO2/HCO3−-free solution, although the increases in pHi and CBF were transient. However, in the presence of acetazolamide, the application of an NH4+ pulse sustained the increases in pHi and CBF. These results suggest that cHNECs may produce a large amount of H+, even under CO2/HCO3−-free conditions.”

  1. It would be of reader convenience to explain why and how application of NH4+ influence the pH within the cell, and what is acetazolamide and the mechanism of its action.
  2. The conclusion in the last phrase is not clear considering the facts presented above. Could it be addressed in more details?

“Thus, cHNECs possess characteristic features distinct from tracheal or lung airway ciliary cells.”

Are there any possible hypothesis of physiological meaning of these differences? Of course, it may be premature to speculate on such subject.

CBD measurement

“Although various abnormal waveforms have been reported in PCD [11], the amplitude of ciliary beating, CBD and CBA, remain controversial as a parameter for PCD diagnosis [12].”

The citation of this work: 12. Neesen, J.; Kirschner, R.; Ochs, M.; Schmiedl, A.; Habermann, B.; Mueller, C.; Holstein, A.F.; Nuesslein, T.; Adham, I.; Engel, W. Disruption of an inner arm dynein heavy chain gene results in asthenozoospermia and reduced ciliary beat frequency. Hum Mol Genet 2001, 10, 1117–1128.“ must be a mistake. This publication concerns the disruption of dynein gene in mouse and does not refer to PCD diagnosis. Moreover, the data presented in this publication show only CBF analysis, no CBD nor CBA were addressed.

Changes in [Cl]i

“[Cl−]i has been shown to modulate cellular functions in many cell types [22–32], and some agonists have been demonstrated to evoke cell shrinkage under isosmotic conditions [20,22,43–46].”

To make this phrase clear, it should be mentioned what type of agonist Authors relating to here. Are these agonists of Cl channels or some other type of agonists?

“The agonists, which stimulate Cl− secretion in airway epithelia, are thought to stimulate KCl release to evoke cell shrinkage.”

Could it be explained in more details why a molecule that increases secretion of Cl- influences also secretion of K+? It is probably trivial for persons working in this field, but must be explained for non-specialists.

“K+ and Cl− are membrane-permeable…”

It would be worth to clarify that these ion are membrane-permeable because of the presence of transport proteins within the membrane.

“For example, assuming that intracellular K+ ([K+]i), [Cl−]i and cell shrinkage are 125 mM, 45 mM and 20%, respectively, the [K+]i is maintained at 125 mM during 20% cell shrinkage. However, a cell shrinkage of 20% causes [Cl−]i to decrease from 45 mM to 25 mM. Thus, the isosmotic cell shrinkage decreases [Cl−]i without any change in [K+]i (Figure 3) [46].”

This reasoning is true for these precise theoretical concentration of ions. But what is known about K+ and Cl- concentration within cHNEC cells before and after cell shrinkage? If the concentration of these ions is different even by 10 mM, then the calculations presented above would give different result. Maybe this reasoning could be omitted. In my opinion, results presented by Authors, that were obtained by treatment of the cHNEC cells with molecules that differ in mechanism of Cl- level regulation clearly indicate and are enough convincing to state that decrease of Cl- is a factor involved in cilia beating regulation.

[Cl−]i regulation of ciliary beating in cHNECs

“The effects of decreased [Cl−]i on CBF and CBD are presented in Figure 5. A switch to a CO2/HCO3−-free solution alone decreased [Cl−]i [9,10].”

The citation of these two references must be a mistake, please cite correct publications.

“Decreasing extracellular Cl− concentration ([Cl−]o) promoted a decrease in F0/F in cHNECs, indicating a decrease in [Cl−]i. Values of F0/F decreased according to [Cl−]os.”

It is not clear why in the first phrase [Cl−]o is used while in the second phrase [Cl−]os is used. What is the difference? Could it be somehow explained in the text?

“The effects of bum, daidzein and CCis on CBF and CBD…”

Because this is the first time in the main text, that the names of these compounds appear (they full names are mentioned only in the legend of Figure 5), please add the full compound names. Also, it would be worth to mention which kind of effect on [Cl-]i they have –are they inhibitors of Cl− entry or activators of Cl− efflux? Addition of this information would be convenient for non-specialists.

[Cl−]i modulation of CBF and CBD in cHNECs

“WNK1 and WNK4 activate NKCC mediated via oxidative stress-responsive kinase/sterile-20-related proline-, alanine-rich kinase phosphorylation (activation) [61,62].”

This phrase is not clear, could it be re-written?, what is NKCC (the abbreviation “NCC” was explained earlier, but not NKCC), what does it mean “NKCC mediated via… phosphorylation”?

“These observations might suggest that WNK1 and WNK4 are physiological [Cl−]i sensors, especially in cHNECs [59].”

This is a very interesting speculation. I would suggest to add few words of discussion concerning the expression of WNTs within respiratory system. From NCBI Gene database it appears that in humans, only WNK1 is expressed at a significant level in lungs. Of course this does not mean that similar pattern of expression will be in the nasal epithelium. Maybe the differences in WNTs expression are one of the reasons, why cHNECs show different response to Cl- level variation compared to airway ciliated cells.

Another issue is the presence of WNTs within ciliary proteomes. I only checked two of them and could not find any of WNT kinases. However, it is possible that they are in cilia at the very low level- such an example are ciliary tip proteins, Cep104, ARMC9 and crescerin (Louka et al., 2018) – of 3 of them only Cep104 appears sometimes in ciliomes. In any case, I think the information on WNT presence/lack in the proteomes of motile cilia should be addressed by Authors.

“To examine the effects of [Cl−]i on CBD and CBF in cHNECs, the relative changes in CBD and CBF (CBD ratio and CBF ratio), which were reported in previous studies [15,16,21], were plotted against the MQAE fluorescence ratio (F0/F, an index of [Cl−]i). CBD (red marks) was inhibited at a lower [Cl−]i than CBF (blue marks).”

For now, Figure 7 is not cited in the main body, it should be cited somewhere here.

“Moreover, the small G-protein Arl13B localised in the microtubule doublet of the axoneme [65]. [Cl−]i modulates G-proteins and inhibits GTPase activity to increase tubulin polymerisation [28–30, 66].”

This information is not related to the regulation of cilia beating frequency and amplitude (cilia frequency and amplitude are not related to the polymerization of tubulin within cilium). Moreover, functions of Arl13B, depending on the organism studied is related to the doublet integrity (mouse) and intraflagellar transport (C. elegans) and not to ciliary beating. Therefore, I suggest to omit this phrases.

Conclusion

“Under physiological conditions, Cl− secretion, which maintains a healthy nasal cavity, induces a decrease in [Cl−]I, which increases CBD in cHNECs.”

Could this idea explained better? It is generally accepted that Cl- secretion within respiratory system is due to the activity of chloride channel expressed in the goblet cells, not in ciliated cells. Of course it does not mean that ciliated cells are deprived of any chloride channels/exchangers etc. that are sensitive to compounds described here. But this issue need more explanation.

Figures

Figure 1.

“(A, B) Video frame images of cHNECs. Panels A and B show the start and the end of the effective stroke, respectively.”

Panel B is signed as “Recovers stroke” is it correct considering that the description claims for “the end of effective stroke”?

Also, it would help to understand these images if the direction of effective stroke was signed and the position of a cilium/cilium tip was signed at the beginning and the end of the effective stroke.

Figure 4.

The images of cells seem to be the same as in Inui et al., 2018. However this is the same Journal, so as I understand there is a permission to re-use them.

Figure 5

“B) Effects of bumetanide (bum), daidzein and carbocisteine (CCis) on CBF, CBD and F0/F ([Cl-]i). Experiments were performed in a CO2/HCO3−-free solution. Bum, daidzein and CCis decreased F0/F, indicating decreases in [Cl−]i. These decreases in [Cl−]i increased CBD, but CBF remained unchanged. (C) Effects of Cl− channel blockers on CBF, CBD and F0/F ([Cl−]i). 5-Nitro-2-(3-phenylpropylamino)benzoic acid (NPPB) and a cystic fibrosis transmembrane conductance regulator inhibitor (CFTR(inh)-172) increased F0/F ([Cl−]is).”

It is not clear why in the last phrase ([Cl]is) is used while in other cases before ([Cl-]i)?

Figure 7

It is not clear why in this Figure Log(F0/F) was used, instead of a simple ration F0/F, as it was used in previous figures. Could the Authors explain this modification?

“In this figure, the plotted F0/Fs, CBD ratios and CBF ratios…”

It is not clear what is the difference between F0/F (used in the majority of the text) and F0/Fs (used exclusively here).

Typos:

5.1. Effects decreased [Cl−]i on CBF and CBD

“of” should be added between “effects” and “decreased”

P12 line 10: WMK3 should be corrected to WNK3

P13 line 7 from bottom:  “Ta.I. M.Y., T.O. and T.N.; funding acquisition” a coma must be added after Ta.I.

Author Response

Thank you for your excellent review.  I have revised our manuscript according to your point out. Please see the revised version of our manuscript which contains modifeied sentences colored red . 

Reviewer 1

The review by Yasuda et al., presents current knowledge on motile cilia beating regulation by intrinsic levels of chloride anion. Presented work summarize mainly achievements of Authors themselves. On the other hand, there are not many other studies in this field. Generally, the manuscript comprises an intriguing topic that would be of interest to readers of IJMS Special Issue on "Molecular Researches on Cilia" and therefore I recommend this manuscript for the publication.

However, before publication I would like to ask Authors to address several issues to improve the manuscript, making the presented ideas more accessible for readers that are not familiar with ion balance within the cells.

Below, find my comments that hopefully will help to improve the manuscript:

Manuscript main body:

A: cHNECs:

A1:      “However, in cHNECs, the switch to a CO2/HCO3−-free solution induced only a small and transient increase in pHi, leading to a slight or sometimes no increase in CBF [15,16,21]. An NH4+ pulse, which induces a large increase in pHi independent of CO2 [21,38,39], significantly increased CBF and pHi in cHNECs, unlike the switch to a CO2/HCO3−-free solution, although the increases in pHi and CBF were transient. However, in the presence of acetazolamide, the application of an NH4+ pulse sustained the increases in pHi and CBF. These results suggest that cHNECs may produce a large amount of H+, even under CO2/HCO3−-free conditions.”

  1. It would be of reader convenience to explain why and how application of NH4+ influence the pH within the cell, and what is acetazolamide and the mechanism of its action.

I have added the following sentence in the last paragraph of page 2.

“For changing pHi, an NH4+ pulse is used for experiments. An application of NH4Cl in extracellular fluid induces a release of a small amount of NH3, which enters cells and traps H+ to produce NH4+ leading to an increase in pHi [21,38,39] In cHNECs, application of the NH4+ pulse induced a larger increase in pHi independent of CO2 than that induced by the CO2/HCO3-free solution, and it significantly increased CBF and pHi, although the increases in pHi and CBF were transient. However, in the presence of acetazolamide (an inhibitor of carbonic anhydrase, which inhibits H+ production (a pHi decrease) from CO2), the application of an NH4+ pulse induced sustained increases, not transient, in pHi and CBF.

  1. The conclusion in the last phrase is not clear considering the facts presented above. Could it be addressed in more details?

I have changed the first part of page 2 and 3, as follows.

“These results indicate that cHNECs produce a large amount of H+, even under CO2/HCO3-free conditions. A high level of H+ production suggests that cHNECs produce a high level of CO2 or the activity of carbonic anhydrase is high in cHNECs. 

A2:      “Thus, cHNECs possess characteristic features distinct from tracheal or lung airway ciliary cells.”

Are there any possible hypothesis of physiological meaning of these differences? Of course, it may be premature to speculate on such subject.

I have added the sentence following to “Thus, cHNECs possess characteristic features distinct from tracheal or lung airway ciliary cells.”. The characteristic features of cHNECs distinct from trachea and lung airway ciliary cells appears to be caused by the different embryological origins. The cHNECs are derived from the surface ectoderm, while ciliary cells of trachea and lung are from the endoderm. These characteristic features appear to be beneficial in cHNECs, which are exposed to air directly. Especially, a high H+ production in cHNECs prevents pHi changes induced by fluctuations in CO2 concentrations during inspiration and expiration (0.3 mmHg – 40 mmHg).

B: CBD measurement

B1:      “Although various abnormal waveforms have been reported in PCD [11], the amplitude of ciliary beating, CBD and CBA, remain controversial as a parameter for PCD diagnosis [12].”

The citation of this work: “12. Neesen, J.; Kirschner, R.; Ochs, M.; Schmiedl, A.; Habermann, B.; Mueller, C.; Holstein, A.F.; Nuesslein, T.; Adham, I.; Engel, W. Disruption of an inner arm dynein heavy chain gene results in asthenozoospermia and reduced ciliary beat frequency. Hum Mol Genet 2001, 10, 1117–1128.“ must be a mistake. This publication concerns the disruption of dynein gene in mouse and does not refer to PCD diagnosis. Moreover, the data presented in this publication show only CBF analysis, no CBD nor CBA were addressed.

Thank you for pointing out the mistake. I am sorry to trouble you in the citation.

I have changed as follows.

“Although various abnormal waveforms have been reported in PCD, the amplitude of ciliary beating, CBD and CBA, remain controversial as a parameter for PCD diagnosis [11].”

C: Changes in [Cl]i

C1:      “[Cl−]i has been shown to modulate cellular functions in many cell types [22–32], and some agonists have been demonstrated to evoke cell shrinkage under isosmotic conditions [20,22,43–46].”

To make this phrase clear, it should be mentioned what type of agonist Authors relating to here. Are these agonists of Cl channels or some other type of agonists?

I have changed as follows.

[Cl]i has been shown to modulate cellular functions in many cell types [22–32] and to be affected by cell volume changes. Many agonists activating ion transport, such as Cl- secretion and K+ release, have been demonstrated to evoke cell shrinkage under the isosmotic condition in many cell types [20,22,43–46].

C2:      “The agonists, which stimulate Cl− secretion in airway epithelia, are thought to stimulate KCl release to evoke cell shrinkage.”

Could it be explained in more details why a molecule that increases secretion of Cl- influences also secretion of K+? It is probably trivial for persons working in this field, but must be explained for non-specialists.

I have added the following sentence in page6

In airway epithelia, an activation of Cl- secretion (Cl- release from cells) also accompanies K+ release for the maintenance of the intracellular electroneutrality. The KCl release, which generates a hypoosmotic condition in intracellular space, induces fluid efflux to evoke cell shrinkage.

C3:      “K+ and Cl− are membrane-permeable…”

It would be worth to clarify that these ion are membrane-permeable because of the presence of transport proteins within the membrane.

See below

C4:      “For example, assuming that intracellular K+ ([K+]i), [Cl−]i and cell shrinkage are 125 mM, 45 mM and 20%, respectively, the [K+]i is maintained at 125 mM during 20% cell shrinkage. However, a cell shrinkage of 20% causes [Cl−]i to decrease from 45 mM to 25 mM. Thus, the isosmotic cell shrinkage decreases [Cl−]i without any change in [K+]i (Figure 3) [46].”

This reasoning is true for these precise theoretical concentration of ions. But what is known about K+ and Cl- concentration within cHNEC cells before and after cell shrinkage? If the concentration of these ions is different even by 10 mM, then the calculations presented above would give different result. Maybe this reasoning could be omitted. In my opinion, results presented by Authors, that were obtained by treatment of the cHNEC cells with molecules that differ in mechanism of Cl- level regulation clearly indicate and are enough convincing to state that decrease of Cl- is a factor involved in cilia beating regulation.

In this section, I would like to explain the mechanism of [Cl-]i decrease evoked by cell shrinkage, using a model cell, not cHNECs. How do cells change [Cl-]i under the physiological condition? For answering this question, we showed a model cell in Fig. 3, to explain how cells decrease [Cl-]i. A cause for [Cl-]I decrease is the cell shrinkage. This is not a familiar idea for many investigators. The reviewer pointed out, that is, if the concentrations of these ions are different even by 10 mM, the results (the calculated value of [Cl-]i) are different. This is true. The result (the calculated value of [Cl-]i) shows the different value. In this model cell, I do not want to show the absolute value. I would like to show how the cell shrinkage does decrease [Cl-]i, but not [K+]i. In this point (a decrease in [Cl-]i), The cell shrinkage does decrease [Cl-]i. This is a model cell, not cHNECs. Unfortunately, we do not measure the exact concentrations of K+ and Cl- in cHNECs. However, in our experiments of cHNECs, the [Cl-]i decrease was coincided with the cell shrinkage. There are many reports showing that the cell shrinkage modulates cellular functions. Although cell shrinkage affects many cellular events, such as an [Ca2+]i increase and channels activation, I think that this modulation is, at least, partly caused by a [Cl-]i decrease.

In our experiments, we measured both cell volume and [Cl-]i (MQAE fluorescence ratio) to strengthen the [Cl-]I decrease, when we can measure the cell volume. We need Fig. 3 for showing an important cause for the [Cl-]i decrease.

 I have changed sentences in page6 as follows.

K+ and Cl, which are the main intracellular cation and anion, are membrane-permeable ions, because cell membrane has ion transporters and channels for K+ and Cl-. The isosmotic cell shrinkage, which is caused by the KCl release, decreases [Cl]i. Figure 3 shows the mechanism for decreasing [Cl-]i induced by cell shrinkage using a model cell for the calculation of [K+]i and [Cl-]i. In this model cell, we assume that intracellular K+ concentration ([K+]i) and [Cl]i are 125 mM and 45 mM in unstimulated condition, respectively. Assuming that an agonist decreases cell volume by 20%, the [K+]i is maintained at 125 mM, but, [Cl]i decreases from 45 mM to 25 mM, as shown in Fig. 3. Thus, in this model cell, the isosmotic cell shrinkage decreases [Cl]i without any change in [K+]i (Fig. 3) [46]. The calculation in this model cell indicates that the [Cl-]i decrease occurs during cell shrinkage under physiological conditions.

D: [Cl−]i regulation of ciliary beating in cHNECs

D1:      “The effects of decreased [Cl−]i on CBF and CBD are presented in Figure 5. A switch to a CO2/HCO3−-free solution alone decreased [Cl−]i [9,10].”

The citation of these two references must be a mistake, please cite correct publications.

I have changed from [9, 10] to [15,16,21] in page 8 line 12.

D2:      “Decreasing extracellular Cl− concentration ([Cl−]o) promoted a decrease in F0/F in cHNECs, indicating a decrease in [Cl−]i. Values of F0/F decreased according to [Cl−]os.”

It is not clear why in the first phrase [Cl−]o is used while in the second phrase [Cl−]os is used. What is the difference? Could it be somehow explained in the text?

I have changed as follows in page 10 line3-5.

“Decreasing extracellular Cl concentration ([Cl]o) promoted a decrease in F0/F in cHNECs, indicating a decrease in [Cl]i. Values of F0/F decreased according to [Cl]o decrease.”

D3:      “The effects of bum, daidzein and CCis on CBF and CBD…”

Because this is the first time in the main text, that the names of these compounds appear (they full names are mentioned only in the legend of Figure 5), please add the full compound names. Also, it would be worth to mention which kind of effect on [Cl-]i they have –are they inhibitors of Cl− entry or activators of Cl− efflux? Addition of this information would be convenient for non-specialists.

 I have changed to “bumetanide (bum, an inhibitor of Na+/K+/2Cl- cotransporter (NKCC)), daidzein (an activator of Cl- channels) and carbocisteine (CCis, an activator of Cl- channels)” in page 10 line 9-10.

E: [Cl−]i modulation of CBF and CBD in cHNECs

E1:      “WNK1 and WNK4 activate NKCC mediated via oxidative stress-responsive kinase/sterile-20-related proline-, alanine-rich kinase phosphorylation (activation) [61,62].”

This phrase is not clear, could it be re-written?, what is NKCC (the abbreviation “NCC” was explained earlier, but not NKCC), what does it mean “NKCC mediated via… phosphorylation”?

I have rewritten as follows in page 12 line20-21; WNK1 and WNK4, which phosphorylate OSR1 (oxidative stress-responsive kinase 1) and SPAK (sterile20-related proline-, alanine-rich kinase), activate NKCC [61,62].

E2:      “These observations might suggest that WNK1 and WNK4 are physiological [Cl−]i sensors, especially in cHNECs [59].”

This is a very interesting speculation. I would suggest to add few words of discussion concerning the expression of WNTs within respiratory system. From NCBI Gene database it appears that in humans, only WNK1 is expressed at a significant level in lungs. Of course this does not mean that similar pattern of expression will be in the nasal epithelium. Maybe the differences in WNTs expression are one of the reasons, why cHNECs show different response to Cl- level variation compared to airway ciliated cells.

I would like to stress that CBF and CBA responses to [Cl-]i increase and decrease are similar in cHNECs and lung airway ciliary cells.

See below.

Another issue is the presence of WNTs within ciliary proteomes. I only checked two of them and could not find any of WNT kinases. However, it is possible that they are in cilia at the very low level- such an example are ciliary tip proteins, Cep104, ARMC9 and crescerin (Louka et al., 2018) – of 3 of them only Cep104 appears sometimes in ciliomes. In any case, I think the information on WNT presence/lack in the proteomes of motile cilia should be addressed by Authors.

In motile cilia of cHNECs, the presence of WNK1 and 4 remains uncertain. But, Olfactory sensory cilia have WNK1 and 4.

 I have added the expression of WNK1 and WNK4 in airway ciliary cells and nasal epithelia. I have changed as follows in page12 line 11-17; “WNK1 and WNK4 have been shown to regulate CFTR [59] and NKCC1 of olfactory sensory cilia [60]. WNK4 has to regulate epithelial Na+ channels (ENaC) in various tissues including airway [61, 62]. CFTR and ENaC exist in nasal epithelia [15, 16, 63]. Moreover, WNK1 and WNK4 have been demonstrated to exist in olfactory sensory cilia [60]. Although the presence of WNK1 and WNK4 in motile cilia of cHNECs still remain uncertain, these observations suggest that WNK1 and WNK4 are physiological [Cl]i sensors in cHNECs, as shown in an NCC study using WNK4 K/O mouse [64]. Further studies are necessary.”

E3:      “To examine the effects of [Cl−]i on CBD and CBF in cHNECs, the relative changes in CBD and CBF (CBD ratio and CBF ratio), which were reported in previous studies [15,16,21], were plotted against the MQAE fluorescence ratio (F0/F, an index of [Cl−]i). CBD (red marks) was inhibited at a lower [Cl−]i than CBF (blue marks).”

For now, Figure 7 is not cited in the main body, it should be cited somewhere here.

 I have added (Figure 7) according to the reviewer’s comment in page 12 line 27. To examine the effects of [Cl]i on CBD and CBF in cHNECs, the relative changes in CBD and CBF (CBD ratio and CBF ratio), which were reported in previous studies [15,16,21], were plotted against the MQAE fluorescence ratio (F0/F, an index of [Cl]i) (Figure 7).

E4:      “Moreover, the small G-protein Arl13B localised in the microtubule doublet of the axoneme [65]. [Cl−]i modulates G-proteins and inhibits GTPase activity to increase tubulin polymerisation [28–30, 66].”

This information is not related to the regulation of cilia beating frequency and amplitude (cilia frequency and amplitude are not related to the polymerization of tubulin within cilium). Moreover, functions of Arl13B, depending on the organism studied is related to the doublet integrity (mouse) and intraflagellar transport (C. elegans) and not to ciliary beating. Therefore, I suggest to omit this phrases.

I have changed to follows in page 12 line 46-50: A previous study showed that a decrease in [Cl]i increased the dynein affinity for microtubules [68]. Fluctuations in [Cl]i, which modulate G proteins and GTPase [28, 30, 69], may also affect the interactions between tubulin and dyneins via G proteins and GTPase, which may change the sliding speed or force generation [70] leading to changes in CBF and CBD. These actions may also be mediated by WNK4 and WNK1 or be independent from them.

F: Conclusion

F1:      “Under physiological conditions, Cl− secretion, which maintains a healthy nasal cavity, induces a decrease in [Cl−]I, which increases CBD in cHNECs.”

Could this idea explained better? It is generally accepted that Cl- secretion within respiratory system is due to the activity of chloride channel expressed in the goblet cells, not in ciliated cells. Of course it does not mean that ciliated cells are deprived of any chloride channels/exchangers etc. that are sensitive to compounds described here. But this issue need more explanation.

I have changed to follows in page 13 line 14- 24; [Cl]i is an important signal ion that modulates ciliary beating in cHNECs. Nasal epithelia in primary culture have already shown to secrete Cl- [63, 71], although nasal mucosal glands including goblet cells also secrete Cl- and mucins maintaining nasal mucous layer. Under physiological conditions, Cl- secretion in nasal epithelia maintains the surface serous fluid layer just below the mucous film, in which cilia beat. Thus, Cl- secretion in cHNECs maintains the mucociliary clearance in nasal and sinonasal epithelia. Our studies in cHNECs demonstrated that an activation of Cl secretion induced a decrease in [Cl]i, which increases CBD.

Figures

Figure 1.

“(A, B) Video frame images of cHNECs. Panels A and B show the start and the end of the effective stroke, respectively.”

Panel B is signed as “Recovers stroke” is it correct considering that the description claims for “the end of effective stroke”?

Also, it would help to understand these images if the direction of effective stroke was signed and the position of a cilium/cilium tip was signed at the beginning and the end of the effective stroke.

I have added the direction of the effective stroke in panel A and that of the recovery stroke in panel B.

Figure 4.

The images of cells seem to be the same as in Inui et al., 2018. However this is the same Journal, so as I understand there is a permission to re-use them.

I agree your opinion. I added the sentence in the last “This figure was modified from Inui et al., 2018 [16] with permission.” I added the outline of cell to show the cell shrinkage.

Figure 5

“B) Effects of bumetanide (bum), daidzein and carbocisteine (CCis) on CBF, CBD and F0/F ([Cl-]i). Experiments were performed in a CO2/HCO3−-free solution. Bum, daidzein and CCis decreased F0/F, indicating decreases in [Cl−]i. These decreases in [Cl−]i increased CBD, but CBF remained unchanged. (C) Effects of Cl− channel blockers on CBF, CBD and F0/F ([Cl−]i). 5-Nitro-2-(3-phenylpropylamino)benzoic acid (NPPB) and a cystic fibrosis transmembrane conductance regulator inhibitor (CFTR(inh)-172) increased F0/F ([Cl−]is).”

It is not clear why in the last phrase ([Cl−]is) is used while in other cases before ([Cl-]i)?

I agree your opinion. I corrected ([Cl-]is) to ([Cl-]i).

Figure 7

It is not clear why in this Figure Log(F0/F) was used, instead of a simple ration F0/F, as it was used in previous figures. Could the Authors explain this modification?

F0/F ia an index of [Cl-]i. I would like to show Cl- concentration response curve of CBF and CBD. In a linear relation, the difference between concentration-response curves of CBF and CBD is not clear (data not shown). However, in the log scale, the difference is clear. Cl- effects on WNK1 and 4 have already been shown in the Log scale, as shown in Terker et al. (2016, ref 57). So we used the log scale.

 “In this figure, the plotted F0/Fs, CBD ratios and CBF ratios…”

It is not clear what is the difference between F0/F (used in the majority of the text) and F0/Fs (used exclusively here).

I agree your opinion. I corrected F0/Fs to Fo/F in this sentence. 

Typos:

5.1. Effects decreased [Cl−]i on CBF and CBD

“of” should be added between “effects” and “decreased”

P12 line 10: WMK3 should be corrected to WNK3

P13 line 7 from bottom:  “Ta.I. M.Y., T.O. and T.N.; funding acquisition” a coma must be added after Ta.I.

I corrected the typos according to your point out.

Reviewer 2 Report

This review combines an extensive summary of available data with an interesting hypothesis about how ciliary beating may be controled by chloride ion sensors. Concepts and experiments are thoroughly explained and figures help for better visualisation making it an informative and useful review about a topic of importance to the field of cilium biology, also for cilia researchers who are not specialists in "ciliary beating".

This review discusses the data on the effects of intracellular Cl- concentration ([Cl-]i) on the beating of cilia in ciliated human nasal epithelial cells (cHNEPC). The authors bring together valuable information on the topic, summarise extensive results of multiple high speed video microscopy observations and experiments and bring forward an interesting hypothesis concerning the mechanism of ciliary beating regulation by [Cl-]i. Thereby, ciliary beating is being defined by two parameters, ciliary beating frequency (CBF) and ciliary beating distance (CBD), an index of the beating amplitude, which are very well explained in the review. In summary, this review describes that a decreased [Cl-]i (induced in different ways depending on the experiments) leads to an increased CBD and increased CBF or not, depending on the temperature. This leads the authors to suggest that [Cl-]i sensors with different affinities for Cl- depending on the temperature be present in the cilia and influence ciliary beating by controlling the activity of the outer and inner dynein arms involved in CBF and CBD respectively. Due to their known inhibition by a range of [Cl-]i corresponding to the range inhibiting CBF and CBD respectively, the authors suggest that two kinases of the WNK family (with-no-lysine kinases), WNK 1 and 4, may be such sensors if they happen to be present in the axoneme.
This review brings a lot of information in a way that is easy to follow and well explained. Each set of data is followed by some explicative sentences including the conclusions drawn and a mechanism for the regulation of ciliary beating by [Cl-] fitting the data is suggested. Additionally, figures display the data in a tangible way which makes the review easy to apprehend by the reader. Thank you for this interesting read.
Minor points/typos:
In the title “5.1. Effects decreased [Cl−]i on CBF and CBD”, “of” is missing (effects of decreased)

In the 5.3 paragraph “a number of reports have suggested that that a decrease in [Cl−]i increases”, “that” is duplicated.

In paragraph 6 (p11), in “The Cl−-biding affinity for the structures controlling the ODAs might”, binding is missing its “n”

Author Response

Thank you for your excellent review. I have changed our manuscript which contains modified sentences colored red according to your point out. 

Minor points/typos:
In the title “5.1. Effects decreased [Cl−]i on CBF and CBD”, “of” is missing (effects of decreased)

Thank you for your point out. I added “of” in the sentence.

In the 5.3 paragraph “a number of reports have suggested that that a decrease in [Cl−]i increases”, “that” is duplicated.

Thank you for your point out. I deleted initial “that” in the sentence.

In paragraph 6 (p11), in “The Cl−-biding affinity for the structures controlling the ODAs might”, binding is missing its “n”

Thank you for your point out. I added “n” in the sentence.

Reviewer 3 Report

The current investigation by Yasuda et al., uses high speed video microscopy to study the effect of chloride ion concentration on cilia beat frequency (CBF) and cilia beat distance (CBD) in human nasal epithelial cells. They report that drop in chloride ion concentration increased CBD and not CBF. On the contrary, an increase in chloride ion concentration suppresses both CBD and CBF. The results of the study are novel and can be useful to the field. However, the authors need to address the following concerns prior to publication.

MAJOR CONCERNS:

  1. They go on to highlight the importance of intracellular chloride concentration on cilia movement, by pharmacological inhibition of chloride channels. However, in many cases such drugs may have non-specific effects. Hence, the authors are requested to repeat the experiments by CRISPR knock-out of the chloride channel. 
  2. In many cases cilia movement in respiratory tract is severely impaired during primary cilia dyskinesia (PCD). PCD often results from mutations in microtubule motor proteins or mutations in other structural proteins of the primary cilia. Is there a possible cross-talk between chloride concentration and proteins mutated in PCD? Authors are requested to address this concern in the discussion section  
  3. Are the effects of chloride ion concentration on cilia movement a global phenomenon, i.e., can similar effects be observed in sperm flagella movement and other tissues where cilia beating is essential? The authors are requested to check the effect of chloride ions on other cell types where cilia beating is essential

Author Response

Thank you for your excellent review. I have changed our manuscript which contains modified sentences colored red accroding to your point out. 

Response:

  1. I used three Cl- channel blockers, NPPB, CFTR(inh)-172 (a CFTR inhibitor) and T16Ainh-A01 (a Ca2+-activated Cl channel inhibitor). These inhibitors may have non-specific effects, as the reviewer pointed out. However, NPPB and CFTR(inh)-172 showed the same effects (CBD and CBF decrease, ref No.15). T16Ainh-A01 (a Ca2+-activated Cl channel inhibitor) has different effects, because of no inhibition of CFTR. I never neglect the possibility that the effects of these three Cl- channel blockers on CBD and CBF are non-specific actions. However, since these blockers increased [Cl-]i, it is certain that increases in [Cl-]i decrease CBF and CBD. The CRISPR knock-out of the chloride channel is an interesting experiment, but, at present, it is difficult to do now. Because, cHNECs are human cells. There are many steps to use siRNA in cHNECs. Unfortunately, we do not have the nasal sample from CF patients, because CF patients are rare in Japan. I think the suggestion of reviewer is the next project. 
  2. I have added following sentence in page 12 line 42-45. “PCD patients have mutations in microtubule motor proteins and other structural proteins controlling ciliary beating. These proteins may have Cl- sensors, such as WNK1and WNK4 and their activities may be affected by [Cl-]i. Their mutation may abolish activations of CBD and CBF by a decrease in [Cl-]i, and may make ciliary activities worse. Further studies are required.”

  3. The Cl- regulation in ciliary beating are not popular. As mentioned in this review, Changes in the amplitude (CBD or CBA) are still controversial even in the airway cilia. I think the effects of [Cl-]I on ciliary beating are a global phenomenon, but no data is available at present. In our preliminary experiments of ependymal ciliary cells from newborn mice, NO3- solution increases CBD, but we did not measure [Cl-]i. Further studies are required.

Reviewer 4 Report

In this review  Yasuda M et al  described how the intracellular Cl concentration regulates the ciliary beating in human nasal epithelial cells, a process necessary for the controlling of the rate of mucociliary clearance. They  summarize the molecular mechanisms of action of Cl concentration on ciliary beating, and described the way of actions of drugs stimulating Cl− secretion, which improve the symptoms of nasal and sinonasal diseases by increasing ciliary beating, and  may be a new therapeutic tool for improving nasal and sinonasal problems. The review is well written and well structured. The only minor issue is that the author could add a schematic representation of the structure of motile cilia  in the introduction section.

Author Response

Thank you for your excellent review. I have changed our manusucript which contains modified sentences colored red according to your point out. 

Response: 

I have added the summary figure (Fig 8) in the conclusion.

I also going to add this figure for the figure abstract.

Round 2

Reviewer 3 Report

The authors have addressed my concerns and the manuscript is ready for publication